# The Impact of Climate Change on Hydroecological Response in Chalk Streams

Annie Visser *, Lindsay Beevers and Sandhya Patidar

Institute for Infrastructure and Environment, School of Energy, Geoscience, Infrastructure and Society, Heriot-Watt University, Edinburgh EH14 4AS, UK; l.beevers@hw.ac.uk (L.B.); s.patidar@hw.ac.uk (S.P.)
* Correspondence: a.visser@hw.ac.uk

**Abstract:** Climate change represents a major threat to lotic freshwater ecosystems and their ability to support the provision of ecosystem services. England's chalk streams are in a poor state of health, with significant concerns regarding their resilience, the ability to adapt, under a changing climate. This paper aims to quantify the effect of climate change on hydroecological response for the River Nar, south-east England. To this end, we apply a coupled hydrological and hydroecological modelling framework, with the UK probabilistic climate projections 2009 (UKCP09) weather generator serving as input (CMIP3 A1B high emissions scenario, 2021 to the end-of-century). The results indicate a minimal change in the long-term mean hydroecological response over this period. In terms of interannual variability, the median hydroecological response is subject to increased uncertainty, whilst lower probability extremes are *virtually certain* to become more homogeneous (assuming a high emissions scenario). A functional matrix, relating species-level macroinvertebrate functional flow preferences to functional food groups reveals that, on the baseline, under extreme conditions, key groups are underrepresented. To date, despite this limited range, the River Nar has been able to adapt to extreme events due to interannual variation. In the future, this variation is greatly reduced, raising real concerns over the resilience of the river ecosystem, and chalk ecosystems more generally, under climate change.

**Keywords:** climate change impact; ecosystem functionality; freshwater ecosystems; UKCP09; hydroecological impact; river health

## 1. Introduction

Under the Convention on Biological Diversity, biodiversity is defined as the variability among living organisms, within & between species and ecosystems [1,2]. Within the public sphere, reasons for preserving biodiversity are, frequently, purely aesthetic, cultural and economic [3]. Critically, the societal cost of biodiversity loss, in terms of ecosystem functionality, may be severe. In recent years, significant progress has been made towards understanding this dependency [2,3]; if not universal, broad consensus points include [4]:

- Increased diversity fosters greater productivity of ecosystem functions;
- The diversity-stability hypothesis [5] states that biodiversity introduces redundancy in the system, thereby introducing both resistance and resilience to environmental change;
- The loss of certain species may have keystone effects which cascade through the ecosystem [6]; for example, Woodward, et al. [7] observed that the presence and absence of freshwater shrimp (*Gammarus pulex*), a dominant predator in chalk streams, exerted a strong influence on detrital processing rates.

Termed the freshwater paradox, freshwaters are disproportionately rich in biodiversity [8]. Rivers and streams cover approximately 0.58% of the world's (nonglacial) surface [9], yet up to 7% of species make freshwaters their home [10,11]. For humans, freshwater is considered the most essential natural resource [12]. In addition to water supply, rivers support prosperity, health, and well-being through the provision of ecosystem services; examples include hydro-hazard regulation, water purification and recreation [13]. Our need for freshwater has seen a rapid decline in freshwater biodiversity; in a 2016 World Wildlife Fund (WWF; see Table A1 for definitions of all abbreviations used) report [14] it was estimated that, between 1970 and 2012, freshwater biodiversity declined by 81%, more than double that of terrestrial and marine combined. Figure 1 illustrates the impact of environmental change on biodiversity, ecosystem functionality and hence the provision of the vital ecosystem services upon which we depend.

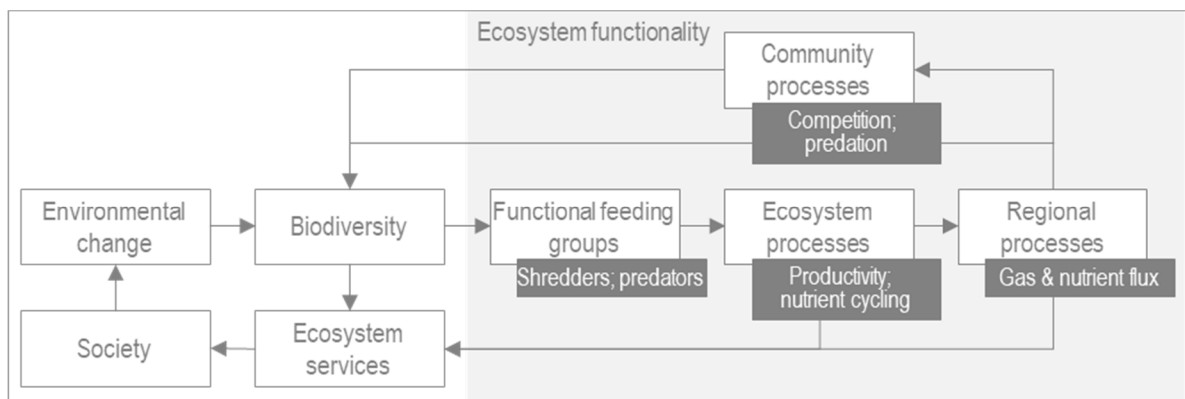

**Figure 1.** Conceptual diagram of the linkages between biodiversity, ecosystem functionality (inset represent examples of functions) and ecosystem services. Based on Chapin et al., 1997 [6] and Cardinale et al., 2012 [4].

The functional composition of the macroinvertebrate community is a major determinant of ecosystem functionality [15]. As consumers at intermediate trophic levels, macroinvertebrates exert strong bottom-up and top-down controls [16]. The above, coupled with their sensitivity to environmental change, makes macroinvertebrates ideal biological and functional indicators [17–19].

Macroinvertebrate functional feeding groups describe their consumption of resources [20], for example, scrapers consume foodstuffs such as algae which are attached to substrate. It is this processing of organic matter which facilitates essential ecosystem processes such as productivity and nutrient cycling [4,6], which in turn supports processes at the regional level. Understanding how the composition of the macroinvertebrate community changes helps to understand the ecological processes in a river, thereby aiding understanding for the purposes of conservation and restoration [21], as well as adaptation to environmental change [22]; the latter being the focus in this study. Flow is widely acknowledged as a major determinant of the health of the river ecosystem (for example, see [23–27]). Data-driven numerical models are used to link flow and hydroecological response in order to understand the instream response to changes in flow [28]. Arguably, the term river health is more useful for interpretation than hydroecological response [29]; hereafter, the term river health should be considered interchangeable with hydroecological response.

Chalk streams provide a steady flow of cool, clear and nutrient-rich water whose gravel channels support uniquely "diverse and fecund ecosystems" [30]. Such streams are famous amongst anglers due to the high levels of fish production that chalk waters are able to support (relative to other river types) [31]. Charles Rangeley-Wilson [30] describes the importance of England's 224 chalk streams as analogous to such biodiversity hotspots as the Great Barrier Reef and equatorial rainforests. Indeed, these streams are (almost entirely) unique to Southern England, with only a handful located in Northern France [30]. The result of a legacy of historical physical modifications—e.g., for systems of

water mills and meadows for irrigation [32] as well as more recent fisheries management [33]—75% of English chalk-streams were designated 'heavily modified water bodies' under the 2008–2012 River Habitat Surveys [30]. Following on from their first report on the state of England's chalk streams a decade prior, the Environment Agency (EA) and WWF-UK concluded that English chalk streams "remain in a shocking state of health" [30,34]. With increasing water demand and climatic variability (e.g., increased hydro-hazards [35,36]), there are significant questions as to the long-term sustainability of this water resource [6,30,37–41]. This is of particular concern given the chalk aquifer provides 70% of the public drinking water in south-east England [30].

The aim of this paper is to quantify the effect of climate change on the river health of a chalk stream. Methods investigating hydroecological response have, typically, been qualitative in nature or quantitative with limited scope, whilst the effect of uncertainty (e.g., parameter, structural, emissions scenario) is rarely considered [42]. To address this research gap, the author's proposed a coupled hydrological and hydroecological modelling framework [42]. The framework was developed using an English chalk stream, the River Nar in Norfolk, where the coupled model was run for a single scenario, CMIP3 SRES A1B high emissions (Coupled Model Intercomparison Project; Special Report on Emissions Scenarios) and 30-year time slice (2041–2070). This paper considers both change in river health over time (from the 2030s to the end of century) as well as the implications for ecosystem functionality. To this end, we consider the same case study river, eliminating the need for model calibration. The UK probabilistic climate projections 2009 (UKCP09) weather generator serves as input to the coupled model; specifically, the high emissions scenario (CMIP3 SRES B1). The results focus on the 99–100% probability, consistent with the Intergovernmental Panel on Climate Change's (IPCC) definition of a virtually certain outcome [43]. The wider implications for chalk streams and groundwater-fed rivers more generally are also reflected upon.

## 2. Case Study Catchment—River Nar

The River Nar, Norfolk, East Anglia (Figure 2) is classified as both chalk and fenland river [44]. For this reason, the river and 180 ha of adjacent land, was designated a Site of Special Scientific Interest (SSSI) in 1992 [33,45], one of only eight chalk streams to be designated as such [30]. In this paper, the focus is on the 24 km chalk river which encompasses an area of 153.3 km$^2$ from the (principal) source at Mileham (TG895194) to the Marham gauging station (TF723119) [46]. Hereafter, all references to the River Nar refer to this chalk reach only.

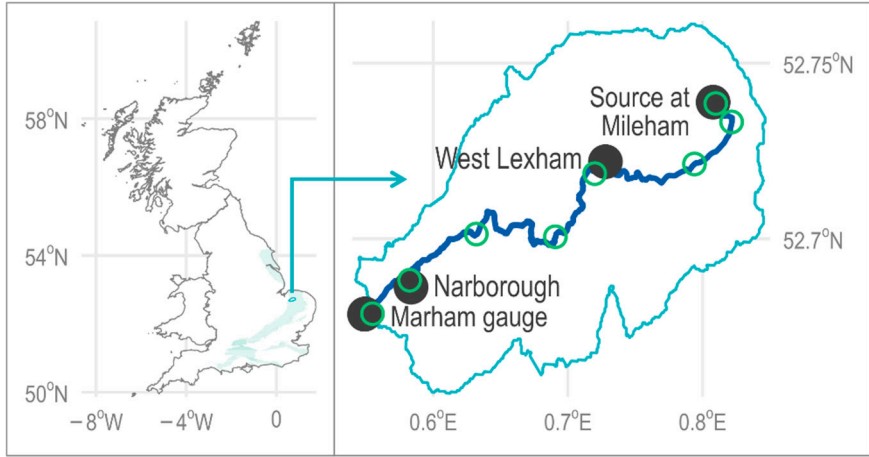

**Figure 2. Left:** Location of English chalk aquifers (shaded) and the case study river catchment (arrow). **Right:** River Nar catchment map; key locations and Environment Agency macroinvertebrate sampling sites are indicated (green).

## 2.1. Hydrology

Flow in the chalk valley is sustained by six springs between West Lexham and Narford Lake (nr. Narborough; Figure 2) [44]. With a baseflow index of 0.91, the hydrology of the river Nar is consistent with that of a classic chalk stream [44]. Typified by a highly seasonal flow regime, aquifer recharge occurs in the autumn months at the start of the hydrologic year (identified as October-November-December; see Figure 3) with flow peaking in January and February (Figure 3). These high flows may see reconnection to floodplain habitats [33]. With a runoff coefficient of 0.35 (1961–1990), flow in the River Nar is indicated as moderately sensitive to change in precipitation [47].

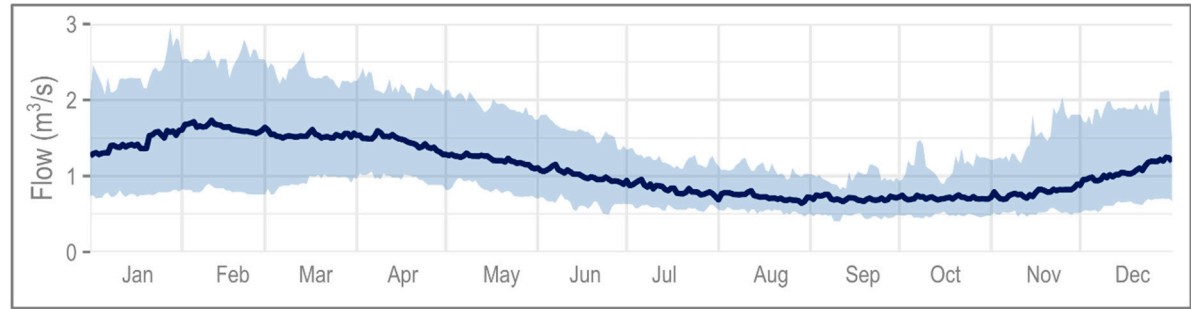

**Figure 3.** Daily median flow recorded at the Marham gauge (1961–1990); the shaded area represents the flow envelope of daily Q90 to Q10 flows. Data source: NRFA, 2018 [46].

## 2.2. Hydrogeomorphological Pressures

The ecological potential of the River Nar is limited to the extent that it is deemed "technically infeasible" for the river to meet the ecological requirements of the Water Framework Directive (WFD) [33,48]. The principle reason is the long history of physical modifications, including Medieval navigation systems, domesday mills, ornamental estate lakes, and most recently, agricultural drainage [33]; only the latter remains functional, providing socio-economic benefits to the catchment. As a low-energy chalk stream, peak flows in the Nar are insufficient to reshape the channel, thus intervention is the only means through which the river might realise its ecological potential. The already fragile state of the river is further exacerbated by sediment ingress as well as over-abstraction for the service of public water supply, fish farms and spray irrigation [30,33,48].

## 2.3. Biodiversity

In chalk streams, peaks in macroinvertebrate activity typically occur in spring (April-May-June where flow begins to recede following winter; hatching season) and autumn (October-November-December; when detritus (food) enters the river system) [32]. Fishing is vital to communities along the River Nar [49,50], as well as chalk streams more generally [30].

Chalk streams are renowned for their abundance of flora and fauna; the high water table and flooding help to support a number of wetland habitats, on the River Nar these include water meadows & pastures, fen wetlands and wet woodlands [49]. From 1993–2017, a total of 188 macroinvertebrate species were observed across 21 orders (see also Table 1); samples were collected by the EA at the eight sites detailed in Figure 2. A total of 12 species of dragonfly *(Odonata)* have been recorded, described as an "outstanding assemblage" in the SSSI designation [49]. Key species such as otters and ecosystem engineers, water voles, have been widely observed in recent years [30,33].

**Table 1.** Number of macroinvertebrate species, grouped by order, observed in the spring season (April-May-June) in the River Nar.

| Order, *Latin Name* (Common Name) | No. Species per Order |
|---|---|
| *Coleoptera* (Beetles) | 35 |
| *Diptera* (True flies) | 3 |
| *Ephemeroptera* (Mayfly) | 17 |
| *Gastropoda c.* (Snails and slugs) | 19 |
| *Hemiptera* (True bugs) | 14 |
| *Odonata* (Dragonfly and damselfly) | 8 |
| *Trichoptera* (Caddisfly) | 52 |
| Other (13 orders) | 40 |
| **Total** | **188** |

## 3. Methods

This paper considers the impact of climate change on river health, hydroecological response, and the implications for ecosystem functionality, in chalk streams. This response is determined through application of a quantitative coupled model [42] with the River Nar serving as case study. Probabilistic climate change projections, from the UKCP09 weather generator, serve as input to the coupled hydrological-hydroecological model. To put this into context, the proxies for river health and ecosystem functionality are first introduced in Section 3.1. An overview of the applied methodology is provided below in Figure 4.

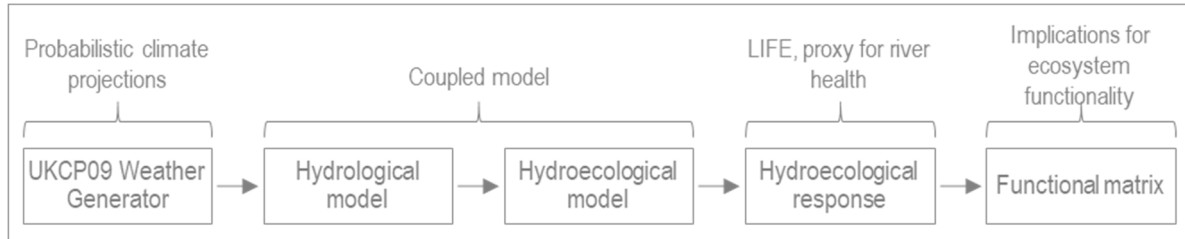

**Figure 4.** Overview of methodological approach.

### 3.1. River Health and Ecosystem Functionality

In this study, the lotic-invertebrate index for flow evaluation (LIFE) [51] serves as the proxy for river health. The LIFE index combines functional flow preferences with the (log) abundance of each taxa to determine flow scores, *fs* (see Appendix A, Figure A1 for a matrix summary of this relationship). The LIFE score is thus determined as:

$$\text{LIFE} = \frac{\sum fs}{n} \tag{1}$$

where the numerator is the sum of the *fs* per taxa, and *n* is the total number of taxa. Lower flow scores, and by extension LIFE scores, are associated with limited flow and standing water, whilst high scores are an indication of rapid flows.

Chapin et al. (1997) [6] stated that no two species are ecologically redundant, it is the diversity within macroinvertebrate functional feeding groups that ensures the resilience of the freshwater ecosystem. Specifically, variation in environmental preferences, such as flow, ensures that a decrease in abundance of one species will be compensated by an increase in a functionally similar species. The importance of diversity, in the context of climate change, and as the freshwater ecosystem responds to more extreme flood and drought events, cannot be understated. A range of represented traits ensures the productivity of the ecosystem.

The impact of pressures, such as climate change, on the functionality of freshwater ecosystems has been limitedly explored, for example [52,53]. Here, we create a matrix of functional flow preferences

and feeding groups (defined in Table A2) using species level macroinvertebrate data collected by the EA at eight sites on the River Nar (Figure 2) from 1993 to 2014 (spring season, April-May-June; $8 * 22 * 1 = 176$ samples) [54]. This 'matrix' highlights which aspects of ecosystem functionality (to date) are most vulnerable to changes in flow. We consider the matrix in the context of the hydroecological projections to elucidate the possible impacts of climate change.

### 3.2. Climate Projections

The UKCP09 probabilistic climate projections, a 25 km grid-square resolution Perturbed Physics Ensemble (HadCM3/HadSM3), serve as the input to the coupled hydrological-hydroecological model. The weather generator was used to produce synthetic stochastic time series at a daily timestep, 5 km grid-square resolution, of climate variables based on observed climate statistics and change factors. The weather generator product was chosen due to its ability to represent climatic variability [55,56], allowing low probability events, vital to ecosystem functionality [57], to be captured more effectively [58]. The climate models upon which the weather generator is based are known for their ineffective simulation of climatic extremes, particularly with regards to precipitation [59]; to address this, the tails of the UKCP09 climate projections are clipped (<5% and >95% probability) [60].

The objective of this study is to explore the change in hydroecological response over time. A range of the CMIP3/SRES scenarios are used in UKCP09: low (B1), medium (A1B) and high (B1); see Figure A2 for scenario specific increase in $CO_2$ emissions. The high emissions scenario was selected due to concerns over the influence that high magnitude change points (Figure A2, highlighted in red) might have on the change signal over time.

Data requests for the required climate variables, precipitation and potential evapotranspiration, were submitted using the UKCP09 web-based portal (http://ukclimateprojections-ui.metoffice.gov.uk/ui/admin/login.php); as of 31 December 2018, data is accessed through the Centre for Environmental Data Analysis (CEDA) archives. The full range of projections (10,000) were considered for each 30-year time slice. As per UKCP09 recommendations, linear bias correction of the climate variables was applied bimonthly (where necessary) [61]. The projections indicate increases in precipitation and potential evapotranspiration in both winter and summer across the three time slices (Figure A3).

### 3.3. Coupled Hydrological-Hydroecological Modelling Framework

The case study river was used by the authors [42] in the development of the coupled hydrological-hydroecological modelling framework. The hydrological and hydroecological models were thus parameterised and validated in the course of the example application, thereby eliminating the need to parameterise and validate the models in this study. To provide context, Sections 3.3.1 and 3.3.2 below provide a brief overview of the hydrological and hydroecological models.

### 3.3.1. Hydrological Model

The four-parameter lumped hydrological model GR4J (Genie Rural a 4 parametres Journalier) [62] was applied using the R package airGR [63]. In summary, the soil moisture accounting model sees: (1) water enter a production store with capacity x1 mm; (2) the water is divided into two flow components, routed through unit hydrographs with time base x4 days; (3) a groundwater exchange term, x2 mm/day, acts upon one component of routed flow, whilst the other enters a routing store with capacity x3 mm; (4) flow in the river is the sum of these two routed flow components.

In the coupled modelling framework, the hydrological model is parameterised using a modified covariance approach which focuses explicitly on the replication of hydrological indicators. Hydrological indicators are used in an effort to improve simulation of the behaviour of the underlying catchment processes [64–66]. Under this approach, the covariance structure of the input (precipitation and potential evapotranspiration) and output (flow) time series are used to identify the region of parameter space which is best able to replicate the characteristics of the hydrological indicators.

The model was parameterised using data over a 54-year period (1961–2015) [42]. The capacity of the production (x1) and routing (x3) stores were estimated at 511 and 311 mm respectively; the time base for flow routing is approximately 1.17 days (x4). A positive groundwater exchange coefficient (x2) of 2.84 mm per day represents inflow from the chalk aquifer.

### 3.3.2. Hydroecological Model

A suite of ecologically relevant hydrological indicators, reflecting Richter's (1996) [67] five facets of the flow regime (magnitude, frequency, duration, timing and rate of change) were considered. In light of seasonality in the flow regime (Figure 3), indicators were determined for both winter (October-November-December-January-February-March) and summer (April-May-June-July-August-September) seasons. Additionally, a one-year time-offset was introduced in order to account for previously observed delays in macroinvertebrate hydroecological response [28,68].

The hydroecological model is developed using multiple linear regression with an information theory approach. This information theory approach provides a measure of the statistical importance of each hydrological indicator (measure of the statistical weight of evidence for the inclusion of the index in the model) in addition to minimising and quantifying uncertainties (structural and parameter). For the structure of the hydroecological model and hydrological indicator definitions, see Equation (A1) and Table A3 in the Appendix A.

### 3.3.3. Analysis

There are no established methods for the analysis due to the relative novelty of the coupled modelling framework [42]. Accordingly, focus fell on the change in distribution of the hydroecological response. For comparison, the projections on the baseline and three future time slices are considered as discrete datasets, with the same methodological approach applied to each. The quantification of uncertainty is central to the application of the coupled modelling framework. To this end, lower and upper bounds of uncertainty where appropriate. Consistent with the IPCC terminology of a *virtually certain* outcome, we use the 99.5% confidence interval [42]. We consider both the aggregated (30-year time slice) and disaggregated (year-on-year) hydroecological response to ensure that the long-term and interannual trends are captured.

## 4. Results

The focus here is on comparison of the distribution of LIFE score, the proxy for river health, over the four time-periods. See Appendix A (Figure A1) for how to interpret LIFE scores relative to functional flow preference. To provide a general overview of the change over time, the long-term trends (aggregate 30-year time slices; Section 4.1) are presented first, followed by the interannual change (Section 4.2) to examine year-on-year variation. Finally, in Section 4.3, the functional matrix, relating functional flow preferences to feeding groups, is considered in the context of these hydroecological projections.

### 4.1. Long-Term Change

The probability density function (PDF; Figure 5) provides a visual representation of the LIFE score distribution for each time slice. The baseline distribution, 1961–1990, sees LIFE scores centered on ~7 (functional flow preference slow to sluggish). From the baseline to the 2030s, the reduction in this clustering coincides with an increase in LIFE scores, whilst the change from the 2030s to 2050s is less marked. The trend for increasing LIFE scores continues into the 2080s where the clustering of LIFE scores can be seen to increase again.

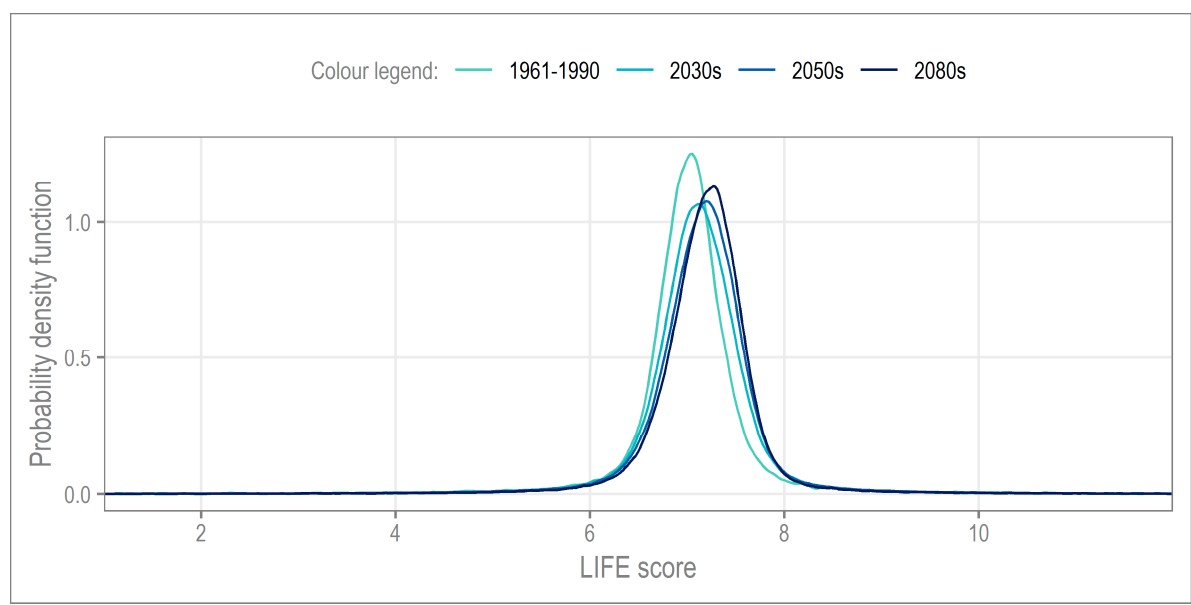

**Figure 5.** Distribution of lotic-invertebrate index for flow evaluation (LIFE; proxy for hydroecological response/river health) projections on the baseline and three futures.

To elucidate further, we consider the standard deviation, as well as the measures of distribution excess kurtosis and skewness (Table 2). The standard deviation reveals an initial increase in variance (2030s), with a subsequent decrease to below the baseline level by the 2080s, suggestive of a slight increase in low probability hydroecological responses by the end of the century. However, the difference across the time slices is relatively small, indicating a limited change in the central distribution of hydroecological response overall.

**Table 2.** Summary statistics of LIFE projections, proxy for hydroecological response or river health.

|  | 1961–1990 | 2030s | 2050s | 2080s |
|---|---|---|---|---|
| Standard deviation | 0.68 | 0.72 | 0.7 | 0.65 |
| Excess kurtosis | 12.43 | 9.75 | 10.57 | 12.9 |
| Skewness | −0.86 | −0.76 | −0.83 | −1 |

Note that for excess kurtosis and skewness, comparisons from baseline to future are not possible, due to differences in sample size (n = 1000 on baseline [69] p. 24). Excess kurtosis is a measure of the combined weight of the tails relative to the normal distribution; for example, a negative value means that more of the dataset is located in the tails than the normal distribution (note that kurtosis is often misinterpreted as a measure of peakedness [70]). Table 2 shows that, for all four time periods, the weight is not located in the tails (hence the observed clustering in Figure 5 previously). Table 2 shows that the change in kurtosis from the 2030s to 2050s, −0.07, is more than half that of the 2050s to 2080s, −0.17. Skewness, a measure of the symmetry in the distribution, shows that all four time-periods are right-skewed; here, the increase from 2050s to 2080s is almost 3 times that of 2030s to 2050s.

In summary, the aggregated projections indicate a very limited change in the mean hydroecological response under climate change. However, Table 2 does highlight that, by the end-of-century, there may be a restructuring of the macroinvertebrate community response to low-probability events. Note that, the smaller scale of change observed between the 2030s to 2050s may be explained by the overlap between these two time slices (2041–2050).

### 4.2. Interannual Variability

The long-term mean may mask significant changes in the interannual variability of hydroecological response. Figure 6 describes vertical cross-sections (at specific quantiles) through annual PDFs of LIFE score; the error bars represent the range of values possible for a *virtually certain* outcome (99.5% probability, based on the available information). Whilst the y-axis for each quantile does vary, it is clear that, perhaps counter to expectation, that the greatest uncertainty surrounds projections of the median response, and across the 5th to 95th quantiles more generally. Next, we consider the change per cross-section (Figure 6), starting with the median, interquartile range (IQR) and finally the tails of the distribution.

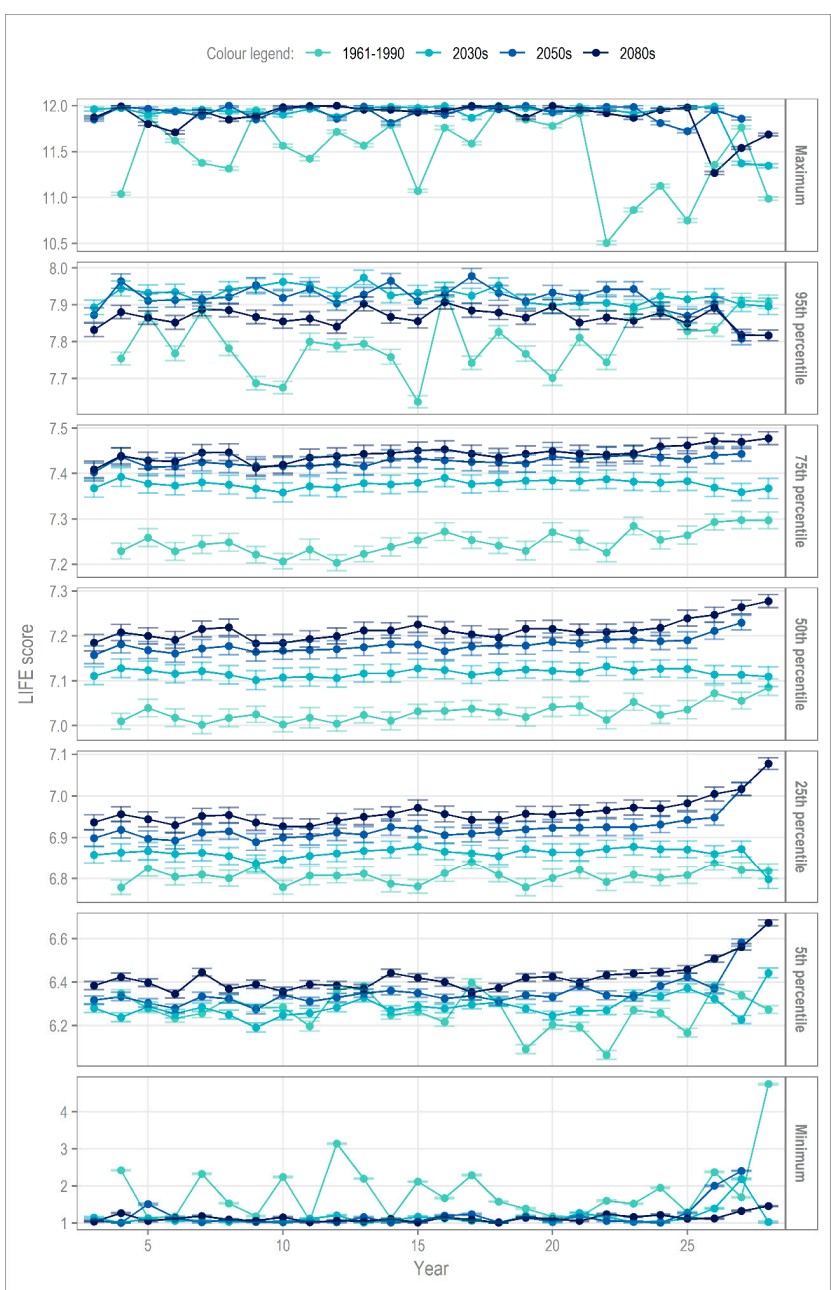

**Figure 6.** Vertical cross-sections (at specific quantiles) through the PDFs of annual LIFE score; the error bars represent the range of *virtually certain* outcomes. The x-axis refers to years 3–28 in each time period (1961–1990, 2030s, 2050s and 2080s), offset is due to the consideration of lag in hydroecological response in the hydroecological model; the y-axis scale is not fixed.

The small shift in median LIFE scores (by time slice) indicates the increased presence of taxa with higher flow scores (Equation (1)), though the variability of LIFE score remains broadly unchanged. At the 75th percentile, a large change occurs between baseline and the 2030s, while the change from the 2050s to 2080s are almost negligible. At the other end of the IQR (25th percentile), the increase in LIFE scores is approximately linear until the end-of the century. As for the 50th percentile, the variation in interannual LIFE scores, per time slice, remains constant.

We now look to the tails of the distribution, essentially the hydroecological response to lower probability extreme events. At the 95th percentile, the change in variance relative to the baseline stands out (Figure 6), with Table 3 revealing that the reduction in variance may reach 92% as early as the 2030s. At the other end of the spectrum, the 5th percentile, the reduction in variance, although reduced, is still high at −65%.

**Table 3.** Percentage change in variance relative to the baseline, per time slice, at the tails of the annual probability density functions (PDFs).

|  | 2030s | 2050s | 2080s |
|---|---|---|---|
| 95th percentile | −92 | −81 | −91 |
| 5th percentile | −65 | −52 | −31 |
| Maximum | −84 | −97 | −83 |
| Minimum | −92 | −84 | −98 |

Changes in the maximum and minimum hydroecological responses are marked, affecting not only variance, but also LIFE score. For the maximum, on the baseline, LIFE scores can be seen to vary from 10.5 to the maximum of 12. However, the projections for all three future time slices show a plateau at LIFE scores of 12; a varied response becomes almost impossible. At the minimum, the same phenomenon is observed, with LIFE scores plateauing to a value of 1 with almost no variance. Further, the reduction in range is more notable than for the maximum.

Examination of the year-on-year change in hydroecological response provides further clarification on the subtle changes observed over the long-term (Section 4.1). Figure 6 and Table 3 also highlight the timing of a major change in hydroecological response could occur as early as the 2030s, 2021–2050. This suggests that a major high or low flow event, in the very near future, could result in a hydroecological response very different to the past (baseline period), where there was the probability of a more varied response. By considering the associated uncertainty, we can be *virtually certain* of this outcome, based on the available information. Given a potentially highly limited period of preparation, this is of concern for the future health of the River Nar.

*4.3. Functional Matrix*

This paper introduces the functional matrix, Figure 7, relating species-level macroinvertebrate functional flow preferences to functional food groups. See Appendix A for definitions. Figure 7 is determined based on observed macroinvertebrate data collected in spring (April-May-June), and thus reflects average conditions between 1993 and 2014. In terms of functional feeding group, only a limited number of species with a range of flow preferences are observed, e.g., scrapers which may tolerate anything from very low to rapid flows. The matrix highlights several functional feeding group traits potentially unrepresented under extreme conditions. The data covers periods of very high and low flows, ensuring that response to extremes are captured. For example, the available time series began in 1993, at the end of the 1989–1992 supra-seasonal drought where groundwater levels fell to their lowest in over 90 years [71]; inadequate groundwater supplies, coupled with increased water abstraction due to the ongoing drought, saw summer and winter Q95 flow fall below 0.16 and 0.19 m$^3$/s, respectively [46].

**Figure 7.** Functional matrix relating functional feeding groups to functional flow preferences at the species level. The values indicate the spring annual average number of species fulfilling a given niche. For example, there are, on average, 2.4 species observed in the upper Nar each spring who fulfill the role of scraper and prefer moderate to fast flows.

In the context of the hydroecological projections, we see an increase in the probability of both very high and low LIFE scores. Looking to Figure A1 in the Appendix A, we can see that LIFE scores below 5/4 are dominated by taxa with preferences for standing waters or drought, and for the highest scores, it is taxa that prefer rapid flows that dominate. Looking then to the functional matrix, it is evident that almost none of the taxa previously observed in the River Nar would be able to perform their functional roles, long-term, under such environmental conditions. In the short-term, the ecosystem has been able to successfully recover, consistent with findings by Wood and Petts [72]. Wood and Petts found in their 1994 study that the impact of drought on chalk streams was, in part, determined by the health of the river ecosystem prior to the drought event. With the projections indicating a reduction in future biodiversity, the concomitant decrease in macroinvertebrate adaptability may significantly impact the resilience of the riverine system.

## 5. Discussion

### 5.1. Impact of Climate Change

Freshwater biodiversity is a major determinant of ecosystem functionality and hence the provision of ecosystem services. Despite this, freshwater biodiversity is declining rapidly around the globe. Coupled with the impact of climate change, there are growing concerns about the long-term sustainability of our water resources.

In this study, we looked at the River Nar, a south-England chalk stream. Using a novel coupled modelling approach, we project how the health of the river may change over time, under a high emissions scenario. The LIFE index served as a proxy for river health. The results showed that, across all three future time slices, interannual variation in LIFE scores is reduced to such an extent that they, essentially, 'flatline'. Over the IQR, the most common hydroecological responses, this change is relatively gradually across the time slices. The change in response at the tails of the distribution is much more marked, with an almost complete loss of variability at both the high and low end of the spectrum by the 2030s.

The overall trend indicates an increased probability and magnitude of extreme responses, with less internal variability. This level of change relative to the baseline conditions has major implications for the structure of the macroinvertebrate community, and hence on ecosystem functionality. The functional matrix, Figure 7, revealed that all functional flow preferences are only met at intermediate flows (LIFE scores range from approximately 6 to 8). Under more extreme conditions, they are effectively 'knocked out'.

To date, the river system has been able to recover from extreme events, indeed, these events may be necessary to ensure the long-term functionality of the ecosystem, acting as a form of "natural reset" [73,74]. However, these responses occurred under a more heterogenous macroinvertebrate community which was adapted to such conditions. With the results indicating a more homogeneous community structure in the future, this may no longer be the case in the very near future. Further, increases in duration of hydro-hazards as reported by Collet et al., 2018 [35] (CMIP3 SRES A1B) and Visser et al., in review [36] (CMIP5 RCP2.6 and RCP8.5) could exacerbate threats to an increasingly vulnerable riverine ecosystem.

## 5.2. Uncertainty

To ensure the validity of the projections, the quantification of uncertainty was central to the application of the coupled modelling framework. To this end, this study utilised the UKCP09 probabilistic climate projections and the UKCP09 weather generator, allowing for the effective capture of lower probability events. To further ensure confidence in the results, a 99.5% probability level was considered. In terms of interannual variability, the bounds of uncertainty are largest for the median and interquartile range, and the greatest confidence lies within the tails of the distribution. Consequently, it is possible to state that a 98% reduction in the variance of hydroecological response by the end of the century is *virtually certain*.

## 5.3. Enhancing and Encouraging Ecological Resilience in Chalk Streams

To our knowledge, this paper represents the first time that quantitative projections of hydroecological response over time have been available. With impacts of climate change being manifest in the river expected as early as 2021 (2030s time slice), the outlook for the River Nar is not promising. A large part of this low resilience may be attributed to the pressures on the river. The River Nar is not alone in this, the State of England's Chalk Streams [30] reporting that, overall, English chalk streams are in a poor state of health, largely for similar reasons. Therefore, whilst this study has focussed on the River Nar specifically, these findings are likely to be more widely applicable to the 200+ chalk streams in England. However, this is not and should not be considered a foregone conclusion, as there remains the opportunity to intervene via improved river management.

Plans for restoration of the River Nar began with the 2010 River Nar Restoration Strategy, with a total of 27 restoration initiatives planned for completion before 2027 [33]. (Note that, in the development of the hydroecological model in Visser et al. [42] and the functional matrix in this study, pre-restoration data was used, 1993–2014.) As the project is completed, and more data is available, this work also presents an opportunity for further study into the effect restoration has on river health and climate change adaptation.

For chalk streams more broadly, a number of positive advancements have been made in recent years. In recognition of the poor condition of chalk streams, there is a drive by Natural England for the reestablishment of a national chalk stream forum [30]; though as of 2018 progress is yet to manifest. The 2014 amendment to the Water Act means that abstraction licence holders on longer have the right to compensation when environmental flow limits are applied. Consequently, water companies are now looking towards investment in measures which ensure water efficiency and thus an overall reduction in abstraction [30].

The fertile chemistry of chalk streams supports their rich ecology and biodiversity, making these systems highly sensitive to changes in nutrient balance. Consequently, management options such as

compensation flows and river transfers are unsuitable in these catchments [49]. A pertinent outcome of the project (EPSRC 1786424), of which this study is part, is the finding that, for the River Nar, up to two years of antecedent flows influence the health of the river; additionally, antecedent winter flows (t-0) are the main determinant of which aspects of the flow regime govern the hydroecological response [42]. See Table A3 for indicator definitions. A summer with high variation in flows could have a significant negative impact on the river two years later; however, a high ratio of Q80 to Q50 flows in the following summer may serve to mitigate these effects. The influence of these antecedent flows become irrelevant when the winter index 10R90Log has either very high or low flow values (dominates LIFE score due to the log nature of the index). These findings indicate a previously unknown degree of flexibility in how the water in the catchment may be utilised. In combination with dynamic environmental flow limits, this represents an opportunity to incorporate with water trading [75,76]. In this way, both the quantity and timing of abstraction may be better managed. Initial scoping studies [75,76] indicated that, for brown trout (*Salmo Trutta*) and mayfly (family *Baetidae*), water trading is unlikely to have a significant impact on habitat availability. However, the study did not consider the importance of this natural variability on the adaptability of the ecosystems or the potential effects of climate change.

## 6. Concluding Remarks

The aim of this paper was to quantify the effect of climate change on hydroecological response in terms of both long-term change and interannual variability. A coupled hydrological and hydroecological modelling framework was forced with UKCP09 high emissions (CMIP3 A1B) projections from 2021 to the end of century. The River Nar, a Norfolk chalk stream, served as the case study catchment. Whilst a minimal change in the long-term mean hydroecological response was projected, the results suggest the homogenization of hydroecological response at the tails of the distribution. At present, the River Nar is shown to be resilient to extreme events despite the absence of key functional groups. With interannual variability contributing to this resilience, the findings in this study raise real concerns over the long-term resilience of the river ecosystem. These new insights into the health of the River Nar, and chalk streams more generally, highlight the necessity of further study and the real need to for changed river management practices. Whilst this work has offered certain pertinent and timely conclusions on the health of the Nar (prior to restoration works), and by extension the chalk stream assemblage across England, it may also be understood as a beginning. The methods are practically applicable across the piece with regards to assessment of the impact of climate change on river health. Further, a better understanding of the River Nar may, and indeed must, facilitate management interventions to safeguard its health and future ecosystem functionality.

**Author Contributions:** A.V. developed the code, performed the data analysis and developed the concept of the functional matrix. A.V. prepared the manuscript whilst L.B. provided review and edits. Both L.B. and S.P. provided supervision.

**Funding:** The authors gratefully acknowledge funding from the Engineering and Physical Science Research Council through award 1786424.

**Conflicts of Interest:** The authors have no conflicts of interest to declare.

## Appendix A

**Table A1.** Definition of the abbreviated terms used in the text.

| Abbreviation | Definition |
|---|---|
| CEDA | Centre for Environmental Data Analysis (UK) |
| CPOM | Coarse particulate organic matter |
| CMIP | Coupled Model Intercomparison Project |
| EA | Environment Agency (UK) |
| FPOM | Fine particulate organic matter |
| GR4J | Genie Rural a 4 parametres Journalier |
| IPCC | Intergovernmental Panel on Climate Change |
| IQR | Interquartile range |
| LIFE | Lotic-invertebrate index for flow evaluation |
| PDF | Probability density function |
| SSSI | Site of Special Scientific Interest (UK) |
| SRES | Special Report on Emissions Scenarios |
| UKCP09 | UK Climate Projections 2009 |
| WFD | Water Framework Directive (EU) |
| WWF | World Wildlife Fund |

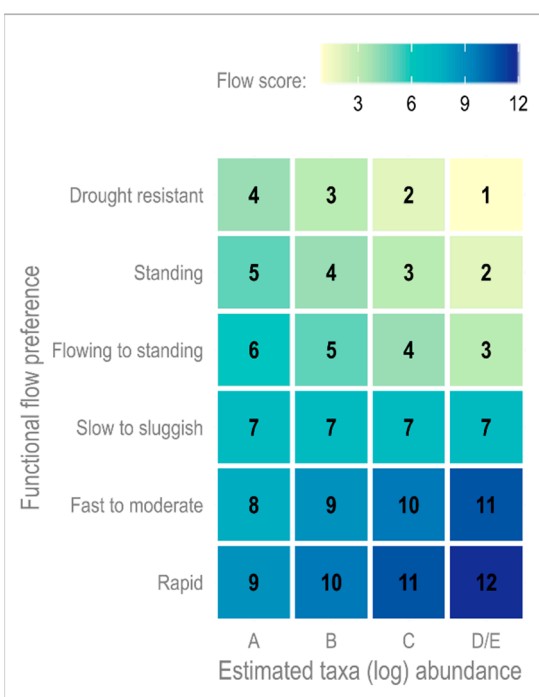

**Figure A1.** Matrix used to determine flow scores *(fs)* in the determination of LIFE scores.

**Table A2.** Description of the seven functional feeding groups considered. Aquatic food resources are classified by size: coarse and fine particulate organic matter (CPOM and FPOM respectively).

| Functional Feeding Group | Description |
|---|---|
| Collector | A broad grouping generally capturing both filterers and gatherers. |
| Filterer | Filter suspended FPOM from the water column. |
| Gatherer | Gather FPOM settled on the substrate. |
| Parasite | Taxa which do not fit into other groups. |
| Predator | Carnivorous macroinvertebrates which prey on smaller invertebrates. |
| Scraper | Consumers of food sources attached to the substrate; e.g., algae and biofilm. |
| Shredder | Shred and consume plant material such as leaf litter and wood. |

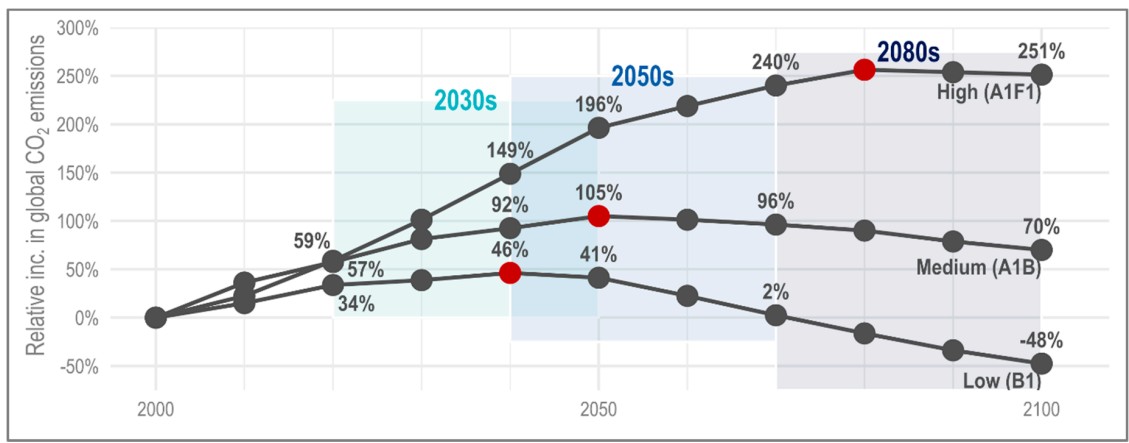

**Figure A2.** The relative (1961–1990 baseline) increase in global $CO_2$ emissions for the three scenarios in UKCP09. Three 30-year time slices are indicated through shading; note that there is some overlap in the 2030s and 2050s slices. Change points, where emissions begin to fall, are indicated in red.

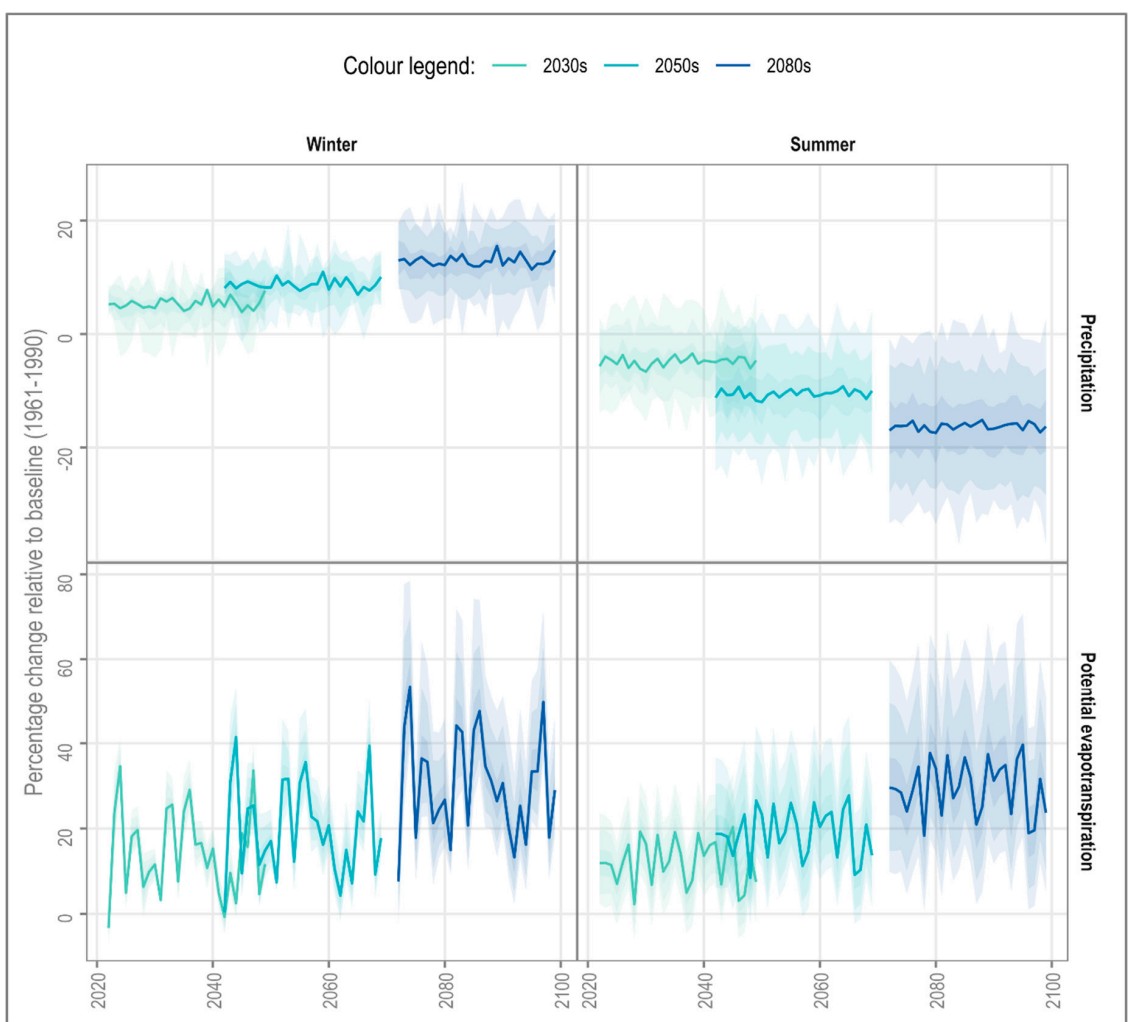

**Figure A3.** Mean percentage change for precipitation (top) and potential evapotranspiration (bottom) across each of the three time periods relative to the 1961–1990 baseline. The solid line indicates the mean, whilst the three envelopes indicate the: interquartile range (darkest), 5th and 95th percentiles (medium) and 1st and 99th percentiles (lightest).

$$\text{LIFE} = 0.07 \ 10R90Log_{w, \ t-0} + 0.07riseMn_{w,t-0} + 0.93 \ Q80Q50_{s, \ t-0} + 0.02 \ Q90Q50_{s, \ t-0}$$
$$+0.3 \ Q90Q50_{s,t-1} + 0.11 \ Q70Q50_{s,t-1} - 0.04 \ RevPos_{s,t-1} - 0.5logQVar_{s,t-1} \tag{A1}$$

**Table A3.** Description of the seven hydrological indicators in the hydroecological model (see Equation (A1)).

| Index Name | Hydrological Season | Time-Offset | Unit | Description |
|---|---|---|---|---|
| $10R90Log_{w,t-0}$ | Winter | t-0 | - | Ratio of log-transformed low to high flows: log(P10)/log(P90). Log-transformation represents the log-normal distribution of flow. |
| $revPos_{s,t-1}$ | Summer | t-1 | days | Number of days when flow is increasing (positive reversals). |
| $Q80Q50_{s,t-0}$ | Summer | t-0 | - | Characterisation of moderate low flows; Q80 relative to the median. |
| $logQVar_{s,t-1}$ | Summer | t-1 | $m^3s^{-1}$ | Variance in log flows. |
| $Q90Q50_{s,t-1}$ | Summer | t-1 | - | Characterisation of low flows; Q90 relative to the median. |
| $Q70Q50_{s,t-1}$ | Summer | t-1 | - | Characterisation of moderate low flows; Q70 relative to the median. |
| $riseMn_{w,t-0}$ | Winter | t-0 | $m^3s^{-1}$ | Mean rise rate in flow. |

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
