# Peer review of "The Impact of Climate Change on Hydroecological Response in Chalk Streams"

_water, doi:10.3390/w11030596_

Reviewer 1 Report

This manuscript shows interesting results on the potential effects of the alteration of flow regime of a chalk stream, originated by climate changes, on the health of this lotic freshwater ecosystem. The manuscript is written properly, but some improvements might be done to facilitate its comprehension in particular for the readers that are not expert in hydroecological modelling. For this reason, I suggest the acceptance of this study after a MINOR REVISION that should covers the following points:

1)      I would prefer a more precise geographical indication of the area of study in the title of the manuscript or in the keywords, as the present manuscript is a case study that analyze only the River Nar (Norfolk, South-England). The generalization of the results can be considered in the Discussion and Conclusions.

 2)      Please, avoid the utilization of the acronyms in the abstract (SSSI; page 1 line 14). In the text of the manuscript, all acronyms (i.e. BFI, AMJ, OND, GR4, ONDJFM, AMJJAS, IQR …) must be defined the first time they appear. To improve the readability of the text, I suggest the inclusion of one table in the Appendix A containing the definition of all acronyms used in this study.

 3)      Page 5 lines 174-176: provide more specific information on the monitoring data that were used in this study (frequency of the sampling, number of data and, if possible, their statistics), as well as on the sampling sites shown in Figure 2. The relevant reference [54] does not indicate the Website Address (URL) where this experimental dataset should be available.

 4)      In the caption of Figure 6, explain better that X-axis refers to the years from 3rd to 28th in each considered period (1961-1990, 2030s, 2050s and 2080s).

 5)      There are few mistakes in the text:

… remain in in a shocking … (page 2 line 82);

… CO2 emissions… : 2 should be a subscript (page 6 line 190, page 13 line 444, Figure A2 Y-axis).

 6)      Figure A2 shows the increases of CO2 emissions estimated by IPCC scenarios that are used in UKCP09 simulations, but the present study is mainly focused on precipitation and potential evapotranspiration trends (page 6 lines 193-198): can the authors show the trends of these parameters in the Appendix A?

Author Response

We thank the reviewer for their feedback and comments. Please find attached a summary table detailing our responses and actions on a comment by comment basis.

Reviewer 2 Report

Please check the attached comments.

Author Response

(The authors gave the same response as above.)

Reviewer 3 Report

Specific comments

 1. The abbreviation “ SSSI ” in the Abstract is difficult to be understand by the reader, please use the words to explain it.

2. The upper mark of “CO2” should be revised into lower mark.

3. Figure 6 the line is not clear, please try to change the style of the figure to present the clear figure to reader.

4. I think the structure should be modified. Putting a brief“Conclusion”part after the “Discussion” maybe much clearer.   

5. The name of figures and tables need to be black. The graduation mark should be clearly and with black color. 

6. The reference format needs to be adjusted into the same format.  

7. Please cite the following references:

Pingping Luo, Meimei Zhou, Hongzhang Deng, Jiqiang Lyu, Wenqiang Cao, Kaoru Takara, Daniel Nover, S. Geoffrey Schladow, Impact of forest maintenance on water shortages: Hydrologic modeling and effects of climate change, Science of the Total Environment, 615, pp. 1355-1363.

Pingping Luo, Dengrui Mu, Han Xue, Thanh Ngo-Duc, Kha Dang-Dinh, Kaoru Takara, Daniel Nover, Geoffrey Schladow,Flood inundation assessment for the Hanoi Central Area, Vietnam under historical and extreme rainfall conditions, Scientific Reports(Nature), 2018, 8:12623, DOI:10.1038/s41598-018-30024-5.

Pingping LUO, APIP, Bin He, Weili Duan, Kaoru Takara, and Daniel Nover: Impact assessment of rainfall scenarios and land-use change on hydrologic response using synthetic Area IDF curves, Journal of Flood Risk Management, Vol.11, pp.S84–S97, DOI: 10.1111/jfr3.12164, 2018.

Pingping LUO, Bin He, Kaoru Takara, Yin E Xiong, Daniel Nover, Weili Duan, and Kensuke Fukushi, Historical Assessment of Chinese and Japanese Flood Management Policies and Implications for Managing Future Floods, Environmental Science & Policy,Vol.48, 2015, pp. 265-277, DOI: 10.1016/j.envsci.2014.12.015.  

Author Response

(The authors gave the same response as above.)
